# Release probability increases towards distal dendrites boosting high-frequency signal transfer in the rodent hippocampus

Thomas P Jensen*, Olga Kopach, James P Reynolds, Leonid P Savtchenko, Dmitri A Rusakov*

Queen Square UCL Institute of Neurology, University College London, London, United Kingdom

**Abstract** Dendritic integration of synaptic inputs involves their increased electrotonic attenuation at distal dendrites, which can be counterbalanced by the increased synaptic receptor density. However, during network activity, the influence of individual synapses depends on their release fidelity, the dendritic distribution of which remains poorly understood. Here, we employed classical optical quantal analyses and a genetically encoded optical glutamate sensor in acute hippocampal slices of rats and mice to monitor glutamate release at CA3-CA1 synapses. We find that their release probability increases with greater distances from the soma. Similar-fidelity synapses tend to group together, whereas release probability shows no trends regarding the branch ends. Simulations with a realistic CA1 pyramidal cell hosting stochastic synapses suggest that the observed trends boost signal transfer fidelity, particularly at higher input frequencies. Because high-frequency bursting has been associated with learning, the release probability pattern we have found may play a key role in memory trace formation.

## Introduction

Information processing in brain circuits relies on dendritic integration of excitatory inputs that occur at varied distances from the soma. Whilst the cable impedance of dendrites attenuates electric signals arriving from distal synapses, some nerve cells appear well equipped to counter this trend. In hippocampal CA1 pyramidal neurons, synaptic receptor numbers increase monotonically with the synapse-soma distance (*Andrasfalvy and Magee, 2001*; *Nicholson et al., 2006*). This, combined with the regenerative properties of local ion channels (*Cash and Yuste, 1999*; *Magee, 1999*), can provide efficient, relatively homogeneous summation of synaptic inputs across the dendritic tree of these cells (*Cash and Yuste, 1999*; *Magee, 2000*; *Magee and Cook, 2000*). In contrast, distal synapses on cortical L5 pyramidal neurons appear relatively ineffective in somatic excitation (*Williams and Stuart, 2002*). A combination of 3D electron microscopy with biophysical modelling of CA1 pyramidal cells has also argued that the greater occurrence of excitatory synapses towards the dendritic branch origin helps to normalise input efficacy along the dendrites (*Katz et al., 2009*). However, it remains largely unknown whether the other fundamental synaptic property, neurotransmitter release probability ($P_r$) remains constant along the dendrites. Because $P_r$ defines how much information synaptic connections transmit over time (*Zador, 1998*), its distribution pattern is critical to the rules of dendritic input integration.

A recent elegant study has combined electron microscopy with extracellular (paired-pulse) afferent activation of multiple synapses in basal dendrites of CA1 pyramidal cells, to find the average $P_r$ values decreasing towards the cell periphery (*Grillo et al., 2018*). Thus, a few proximal connections appear to have a greater impact on the cell output than more numerous synapses occurring distally. This observation raised the question of the synaptic connectome efficiency, suggesting that, during

*For correspondence:
t.jensen@ucl.ac.uk (TPJ);
d.rusakov@ucl.ac.uk (DAR)

Competing interests: The authors declare that no competing interests exist.

continued afferent spiking, distal inputs are weakened. Although highly supra-linear local events, such as dendritic spikes could, in principle, compensate for this trend (*Larkum and Nevian, 2008*; *Branco and Häusser, 2009*), how prevalent such events are remains debatable. Furthermore, if $P_r$ tends to decrease towards more distal apical dendrites in CA1 pyramidal cells, the suggested input summation rules in these cells (*Magee, 2000*; *Magee and Cook, 2000*; *Branco and Häusser, 2011*), and at least in some other principal neurons (*Araya et al., 2006*), would be broken during sustained network activity.

Methodologically, paired-pulse recordings of multi-synaptic responses (*Grillo et al., 2018*), while revealing the trend, do not provide direct readout of either $P_r$, paired-pulse ratios (PPRs), or the $P_r$ distribution among individual connections. This makes it difficult to evaluate how these properties of $P_r$ affect dendritic integration. To address these challenges, here we monitored $P_r$ in individual CA3-CA1 synapses, by systematically employing single-synapse optical quantal analyses (OQA) (*Oertner et al., 2002*; *Emptage et al., 2003*), based on $Ca^{2+}$ imaging in individual dendritic spines, as established previously (*Sylantyev et al., 2013*; *Boddum et al., 2016*). In a complementary approach, we employed the genetically encoded optical glutamate sensor iGluSnFR, as established earlier (*Jensen et al., 2017*; *Jensen et al., 2019*; *Henneberger et al., 2020*; *Kopach et al., 2020*), to document a spatial trend of $P_r$ values across the *s. radiatum*. Exploring the results with a realistic biophysical model of the CA1 pyramidal cell (*Migliore et al., 1999*) equipped with multiple stochastic synaptic inputs, revealed how the observed trends in the $P_r$ pattern along dendrites could affect synaptic input integration in CA1 pyramidal cells.

## Results

### Monitoring release probability with optical quantal analysis

We used 300–350 µm transverse hippocampal slices of 3- to 4-week-old rats (Materials and methods). First, we held CA1 pyramidal cells in whole-cell ($V_m$ = −65 mV) and dialysed them with the red morphological tracer Alexa Fluor 594 (50 µM) and $Ca^{2+}$-sensitive indicators as detailed in Materials and methods and described previously (*Sylantyev et al., 2013*; *Zheng et al., 2015*). We visualised cell morphology with two-photon excitation ($\lambda_x^{2P}$ = 800 nm), and placed the extracellular stimulating pipette in the 10–20 µm proximity of apical dendrites (*Figure 1A*, *Figure 1—figure supplement 1A*). The system was focused on individual dendritic spines that responded to paired-pulse stimuli (50 ms apart) with localised $Ca^{2+}$ transients in a stochastic manner (*Figure 1B*, *Figure 1—figure supplement 1B*). Because individual dendritic spines of CA1 pyramidal cells host almost exclusively only one CA3-CA1 synapse (*Harris and Stevens, 1989*; *Bloss et al., 2018*), the first-pulse $Ca^{2+}$ signal success counts over 14–30 trials provided direct readout of $P_r$ at individual synapses (P1 in *Figure 1B*, *Figure 1—figure supplement 1C*), as we showed earlier (*Sylantyev et al., 2013*; *Boddum et al., 2016*). In these tests, the amplitudes of successful first-pulse release signals were consistently above two standard deviations of the average signal during release failures (*Figure 1—figure supplement 1D*), enabling us to readily calculate $P_r$ = P1. The second-pulse release probability was estimated using two complementary methods (*Figure 2B*): one in which all trials were assessed for 1st and 2nd release successes and failures (P2), and one in which trials with the first successful response were ignored (P2*). The latter method eliminates false-positive detection of the second-pulse success, at the expense of reduced trial numbers. The curvilinear distance from the spine of interest to the cell soma was subsequently recorded (Materials and methods).

A systematic application of this protocol at n = 67 individual synapses (53 cells) has revealed a highly significant trend towards higher $P_r$ at more distal synapses (*Figure 1C*), with average $P_r$ = 0.36 ± 0.02 (mean ± SEM). The linear regression for the data suggested a rise in average $P_r$ from ~0.2 at 50 µm to ~0.5 at 300 µm (range 0.05–0.7; *Figure 1C*). Over this distance range, the PPR of release probabilities decreased, on average, from ~3 to ~1.1, for either P2 or P2* counts (*Figure 1D–E*; experiments with no reliable 2nd response detection and no 'failure-success' responses were not included). While these relationships show relatively low values of correlation coefficient, the interpretation of Pearson's r that is commonly adopted for a specimen-centred design is irrelevant in the present context (see Discussion).

There observations appear at variance with an elegant earlier report which detected no distance-dependent $P_r$ trends along apical dendrites of CA1 pyramidal cells, based on the use-dependent

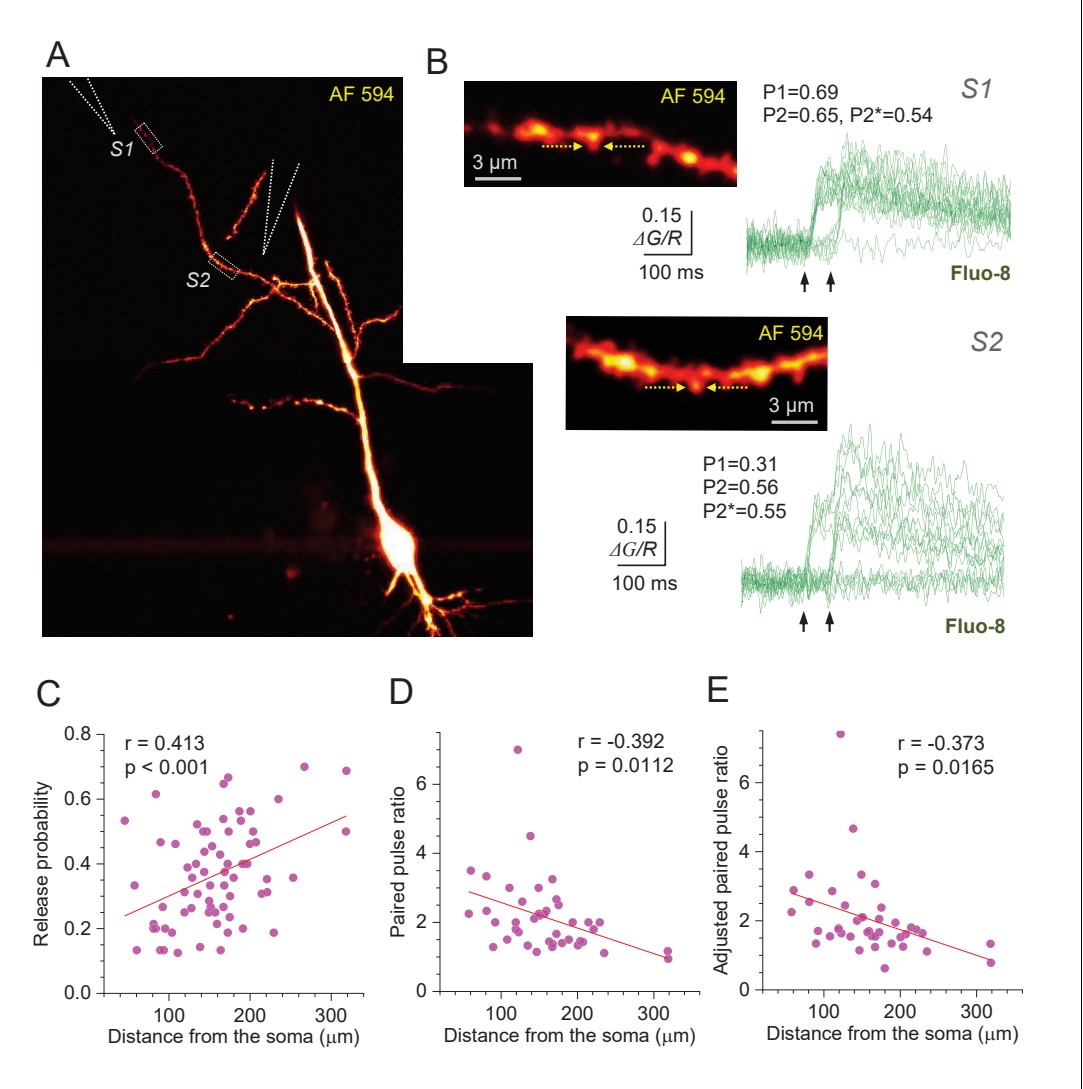

**Figure 1.** Optical quantal analysis at individual CA3-CA1 synapses reports higher release probability towards distant dendrites. (**A**) CA1 pyramidal cell held in whole-cell mode (acute hippocampal slice), dialysed with 50 μM AF 594 and 300 μM Fluo-8 (75 μm z-stack average, $\lambda_x^{2p}$ = 800 nm; AF 594 channel). Dotted rectangles (**S1** and **S2**), two ROIs to record from two dendritic spines; stimulating electrode positions are illustrated by dotted cones. (**B**) Image panels, ROIs as in (**A**), shown at higher magnification; arrows, linescan positioning at the dendritic spines of interest. Traces, Ca$^{2+}$ signal (ΔG/R, Fluo-8 green-channel signal ΔG related to red-channel AF 594 signal R) recorded as the width-integrated linescan intensity, at two spines as indicated, in response to paired-pulse afferent stimuli (arrows). Release failures and responses to the first pulse can be clearly separated (***Figure 1—figure supplement 1D***). P1 and P2 show average release probability in response to the 1st and 2nd stimulus, respectively, calculated as P1 = {(1,0) + (1,1)} / N and P2 = {(0,1) + (1,1)} / N, and P2* is the adjusted 2nd-stimulus release probability calculated as P2* = (0,1) / {(0,0) + (0,1)} where brackets indicate the counts of paired-pulse successes (1) or failures (0), and N is the number of trials. (**C**) Average release probability (P$_r$; shown as P1 in B) at individual synapses plotted against distance to the soma. Solid line, linear regression (p value to reject H$_0$ = zero slope and Pearson's r shown; n = 67). (**D**) Paired-pulse ratio P2/P1 plotted against distance to the soma. Other notations as in (**C**) (n = 41; average PPR mean ± SEM: 2.14 ± 0.17; spines with no reliable detection of 2nd responses, and no detectable (0,1) responses, were excluded). (**E**) Adjusted paired-pulse ratio PPR*=P2*/P1 plotted against distance to the soma. Other notations as in (**C**) (n = 41; average PPR*=2.05 ± 0.18).

The online version of this article includes the following source data and figure supplement(s) for figure 1:

**Source data 1.** Original data readout for ***Figure 1C-E*** and ***Figure 1—figure supplement 1D***.

**Figure supplement 1.** Optical quantal analysis at individual CA3-CA1 synapses in acute hippocampal slices: second-order dendrite.

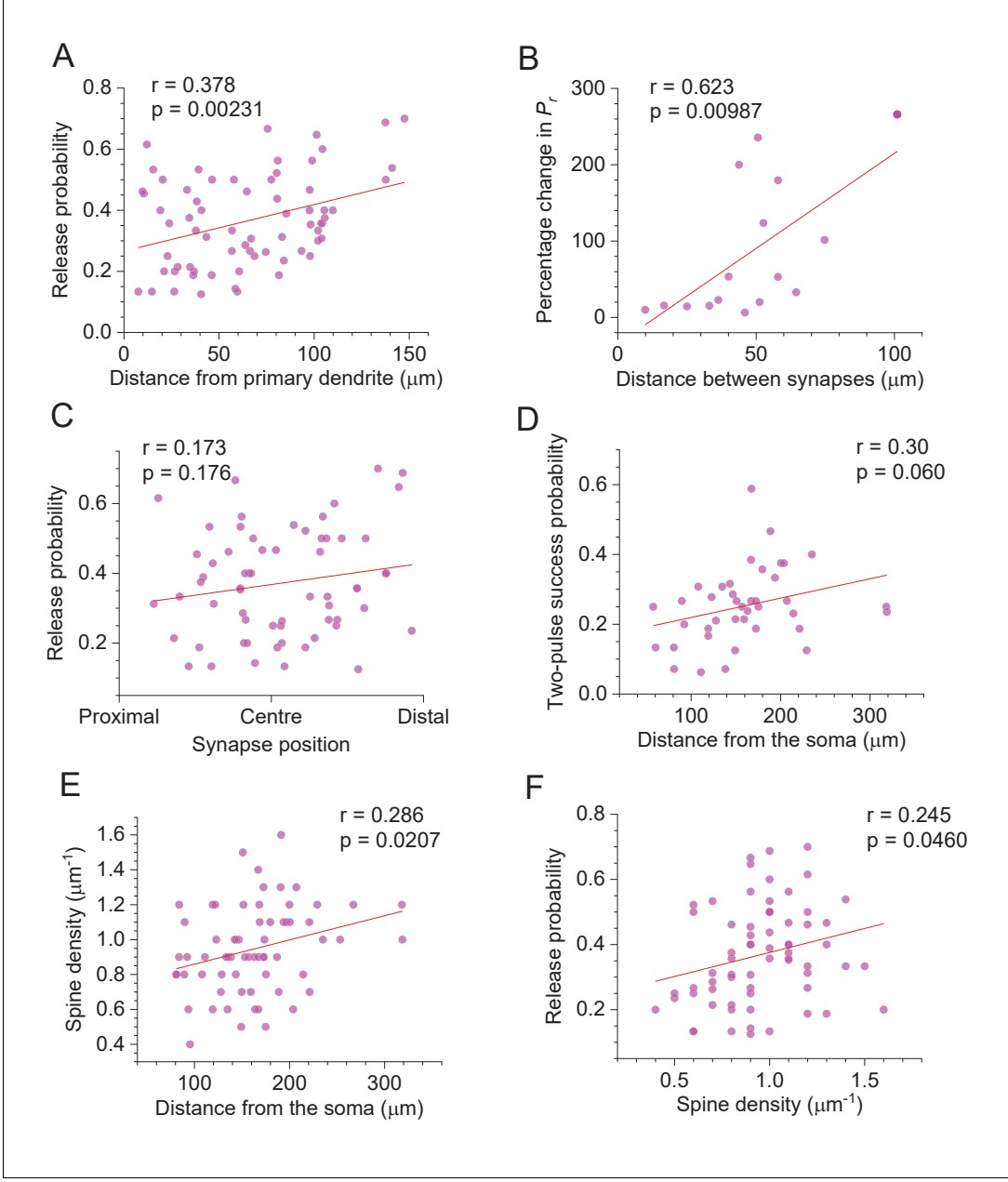

**Figure 2.** Selected features of excitatory synapses with respect to their dendritic location. (A) Average release probability ($P_r$) at individual synapses plotted against distance to the first dendrite branching point. Solid line, linear regression (p value to reject $H_0$ = zero slope and Pearson's r shown; n = 67). (B) Percentage difference in $P_r$ between two synapses on one dendritic branch (as in *Figure 1A*, *Figure 1—figure supplement 1A*), plotted against the distance between them along the branch. Other notations as in (A) (n = 15). (C) Average release probability ($P_r$) plotted against relative synapse position at the dendritic branch: synapse co-ordinate was scaled to the 0–1 range representing the branch origin and the end, as indicated (n = 63; several spines with unidentifiable distal branch ends were excluded). (D) Probability of release success upon both afferent stimuli, plotted against distance from the soma. Other notations as in (A) (n = 40). (E) Apparent spine density along the dendrite (smallest/thinnest spines could be undetectable, see Discussion), plotted against distance from the soma. Other notations as in (A) (n = 65; several spines with unidentifiable local spine density were excluded). (F) Release probability ($P_r$), plotted against distance from the soma. Other notations as in (A) (n = 67).

The online version of this article includes the following source data and figure supplement(s) for figure 2:

**Source data 1.** Original data readout for *Figure 2A-F* and *Figure 2—figure supplement 1A-B*.
**Figure supplement 1.** Selected features of excitatory synapses with respect to their dendritic location.

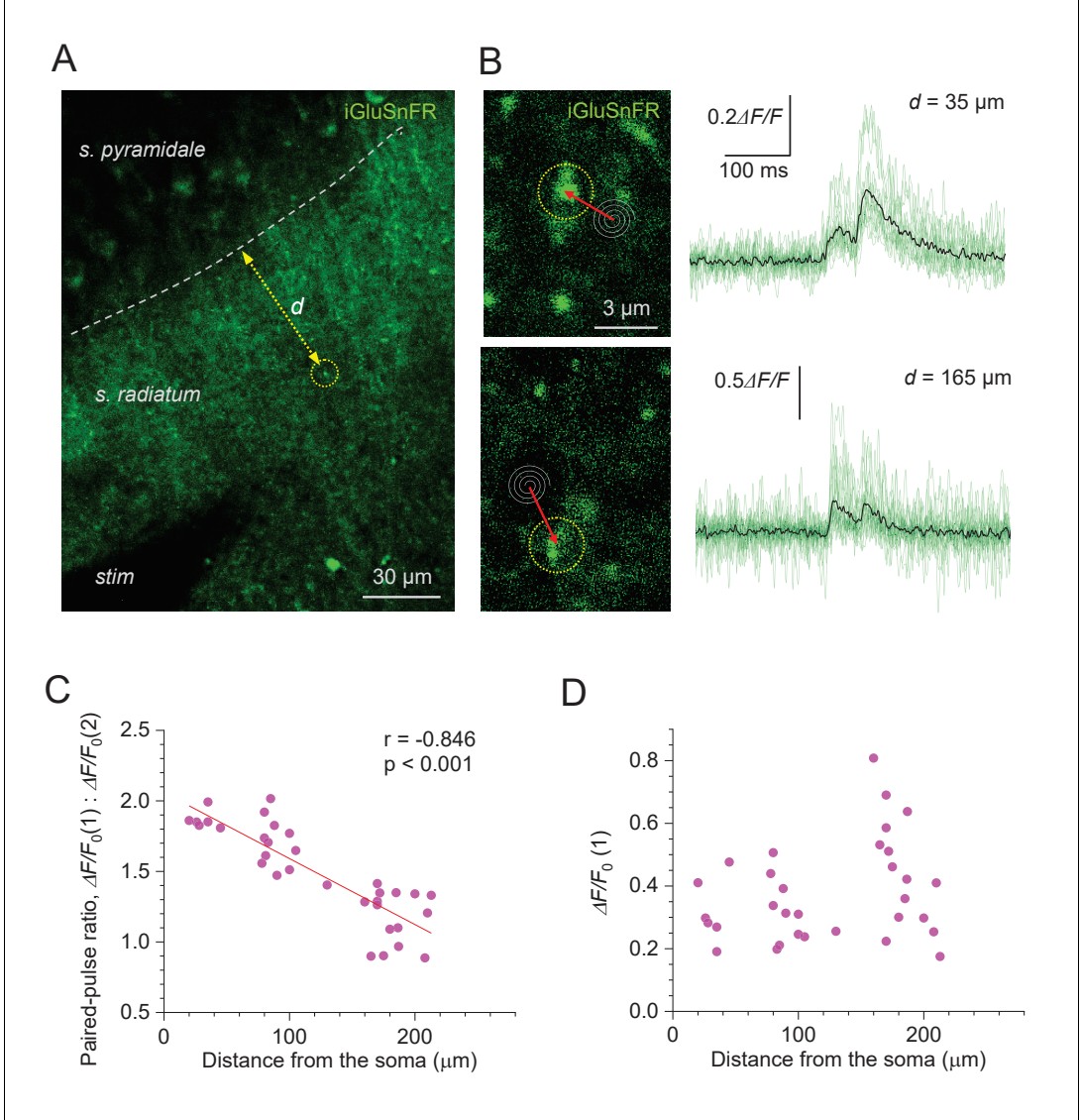

**Figure 3.** Evoked glutamate release from Schaffer collaterals shows lower paired-pulse ratios at greater distances from pyramidal cell bodies. (A) Experimental design: area of the hippocampal slice with iGluSnFR expressed in neuronal membranes (green channel); arrow, measured distance $d$ between CA1 pyramidal cell body layer and the axonal bouton of interest; stim, stimulating electrode. (B) Examples of recorded axonal boutons (image panels, dotted circles; position of spiral 'Tornado' linescans is illustrated) showing characteristic glutamate signals in response to two afferent stimuli 50 ms apart (green traces, individual trials; black, average), at two distances from the s. pyramidale, as indicated. The lack of release failures reflects detection of glutamate escaping from multiple neighbouring synapses. (C) Paired-pulse ratio for optical glutamate signals: $\Delta F/F_0(1) / \Delta F/F_0(2)$ averaged over 18–36 trials at individual boutons, plotted against distance to the soma. Other notations as in *Figure 1C* (n = 33). (D) Amplitude of the first glutamate response, $\Delta F/F_0(1)$ averaged over 18–36 trials at individual boutons, plotted against distance to the soma. The amplitude values reflect the average amount of glutamate released from the bouton of interest, and glutamate escaping from its neighbours; other notations as in (C) (n = 33). The online version of this article includes the following source data and figure supplement(s) for figure 3:

**Source data 1.** Original data readout for *Figure 3C-D* and *Figure 3—figure supplement 1C*.

**Figure supplement 1.** Optical (multi-synaptic) glutamate signal evoked by short bursts of Schaffer collateral stimulation at varied distances from *s. pyramidale*.

blockade of NMDARs by MK801 (*Smith et al., 2003*). However, the MK801-based approach is sensitive to local glutamate spillover (*Scimemi et al., 2004*) and, in that case, involved a $Mg^{2+}$-free solution, both of which could mask the $P_r$ trend. Intriguingly, that study documented 15–50% lower average PPR values in distal compared to proximal synapses, consistently over 25–200 ms paired-

pulse intervals, albeit without reaching statistical significance in a six-cell sample (*Smith et al., 2003*).

The centrifugal trend in $P_r$ values remained significant when the distance was measured from the primary apical dendrites (*Figure 2A*), suggesting that this trend applies to oblique branches rather than simply reflecting a position along the main dendrite. Interestingly, the analysis of $P_r$ values recorded in pairs of synapses located on the same dendrite has revealed that the connections with similar $P_r$ tend to occur close to one another (*Figure 2B*). Thus, 'homogenisation' of $P_r$ among local synapses, which was detected previously in cultured neurons (*Branco et al., 2008*), is likely to occur in ex vivo brain tissue. At the same time, we found no trends in $P_r$ values with respect to the synapse position relative to the local dendrite branching point (*Figure 2C*; the centrifugal $P_r$ trend in these data does not reach statistical significance because branch length is much smaller than the overall dendritic length, thus providing much reduced statistical power). The prevalence of larger, more densely packed spines near the dendritic branch origins was earlier reported to reflect normalisation of synaptic signals integrated at the soma (*Katz et al., 2009*). In this context, no gradient in $P_r$ values within a branch (*Figure 2C*) suggests that the above normalisation should still be valid during sustained afferent activity.

Because higher $P_r$ are normally associated with lower PPRs (as in *Figure 1D*), the probability of both stimuli initiating successful release showed only a barely detectable increase with the spine-soma distance (*Figure 2D*), whereas the second-release $P_r$ values were evenly spread along the dendrites (*Figure 2—figure supplement 1A*). However, the longitudinal density of optically identifiable dendritic spines increased towards distal dendrites (*Figure 2E*), also showing positive correlation with the $P_r$ (*Figure 2F*), but no correlation with the PPR values (*Figure 2—figure supplement 1B*).

## Monitoring glutamate release at CA3-CA1 synapses with an optical sensor

We have previously shown (*Jensen et al., 2017*; *Jensen et al., 2019*) that the patterns of release probability at CA3-CA1 synapses can be successfully gauged by monitoring presynaptic glutamate release with the genetically encoded optical sensor iGluSnFR (*Marvin et al., 2018*). While the earlier work focused on individual axonal boutons tracked from one presynaptic cell, here we attempted to employ a similar method using local extracellular stimulation. Once iGluSnFR (AAV9.hSynap. iGluSnFR. WPRE.SV40) was expressed in the area CA1 *s.radiatum,* we therefore focused on individual presynaptic boutons that showed a fluorescence response to paired-pulse stimuli applied locally to Schaffer collaterals; the distance between the bouton and the *s. pyramidale* border was documented (*Figure 3A*). We thus recorded fluorescence responses to paired-pulse stimuli (50 ms apart) using a high-resolution spiral linescan ('Tornado') mode, as described earlier (*Jensen et al., 2017*; *Jensen et al., 2019*; *Figure 3B*, image panels).

It turned out that in such recordings we could not reliably detect failures, throughout the trials (examples in *Figure 3B*). This has a simple explanation. In the hippocampal neuropil, released glutamate molecules can travel for up to ~2 µm from the synapse (*Armbruster et al., 2020*), not necessarily entering other synaptic clefts (see Discussion). Given the density of glutamatergic connections in area CA1 of ~2.0 µm$^{-2}$ (*Rusakov and Kullmann, 1998*), individual synapses have >60 neighbours within the 2 µm proximity (i.e. within the spherical volume of ~33 µm$^3$). Furthermore, two-photon excitation imaging collects emission signal integrated over a ~ 1-µm-thick focal plane. Thus, extracellular stimulation is likely to generate an iGluSnFR fluorescence transient due to glutamate spillover even if only a small fraction of local axons gets excited. This implies that the iGluSnFR signals reported here reflect release activity of several local synapses, which makes this approach suitable for PPR measurements rather than for direct $P_r$ estimates.

Measuring the amplitudes of paired-pulse iGluSnFR signal transients thus revealed a clear trend towards lower PPR, hence higher $P_r$ values, with greater distances from the *s. pyramidale* (PPR mean ± SEM: 1.49 ± 0.01, n = 33; regression at p<0.001; *Figure 3C*). This observation was qualitatively consistent with the PPR data obtained with the OQA of $Ca^{2+}$ signals (*Figure 1D-E*) . In contrast, the first iGluSnFR response amplitude on its own showed no distance dependence ($\Delta F/F_0$, mean ± SEM: 37 ± 0.5%; range 0.18–0.81; *Figure 3D*). The latter was expected because, in addition to $P_r$ per se, the iGluSnFR signal amplitude depends on several poorly controlled local concomitants, such as the degree of glutamate spillover (which in turn depends on the local expression of iGluSnFR and glutamate transporters) and/or density of activated axons.

In a complementary approach, we recorded iGluSnFR responses to a short burst of five afferent stimuli (at 20 Hz), in an attempt to relate their use-dependent release properties to their location with respect to the *s. pyramidale*. To optimise ROI selection under burst stimulation, these recordings were carried out in a frame (time-lapse) mode, as detailed previously (*Henneberger et al., 2020*; *Kopach et al., 2020*; *Figure 3—figure supplement 1A*). While such recordings have relatively low temporal resolution, the underlying (fast) fluorescence kinetics could be reconstructed using a straightforward fitting procedure (Materials and methods; *Figure 3—figure supplement 1B*). We used the slope of linear regression among the five $\Delta F/F_0$ peak points that correspond to the five stimuli onsets (*Figure 3—figure supplement 1B*) as a crude indicator of short-term release plasticity during the burst. Intriguingly, the slope values increased significantly with greater distances to the *s. pyramidale* (*Figure 3—figure supplement 1C*), suggesting greater fidelity in signal transfer by spike bursts towards more distal dendrites, as further explored below.

## A realistic biophysical model explains the role of the release probability trend

To understand how the uneven distribution of $P_r$ values affects signal handling by the postsynaptic cell, an earlier study used simulations with a sphere-and-cylinder cell model (*Grillo et al., 2018*). Here, we employed a realistic multi-compartment model of a reconstructed CA1 pyramidal cell (*Migliore et al., 1999*), with 50 excitatory synapses distributed along apical dendrites, so that their positions, density, and $P_r$ could be set as required (*Figure 4A*, Materials and methods). In the first test, we asked whether and how the documented centrifugal increase in $P_r$ and synaptic density (termed '$P_r$ trend' and 'density trend', respectively) changes cell output under a synchronous discharge of Schaffer collaterals. A series of 100 paired-pulse stimulus test runs was simulated, each generating stochastic 'glutamate release' at the 50 synapses, either with the same average $P_r$ (0.36) at uniformly scattered synapses, or with the $P_r$ values distributed along the regression line (*Figure 1C*), or, additionally, with the synaptic density trend as found (*Figure 2E*), with the average $P_r = 0.36$ kept unchanged throughout, to ensure the unchanged efficacy of the overall synaptic input.

The results show that the $P_r$ trend, on average, adds ~13% to the single-pulse EPSP amplitude while decreasing PPR by ~20%, compared to the evenly distributed $P_r$, whereas adding the density trend produces little further change (*Figure 4A–B*). Qualitatively similar results were obtained for the five-pulse burst tests based on the iGluSnFR data, suggesting that the $P_r$ trend, when combined with the five-pulse slope trend (*Figure 3—figure supplement 1C*), boosts the voltage transfer value during the burst activity by ~15% (*Figure 4—figure supplement 1A–B*).

However, the main role of $P_r$ in synaptic signal integration and transfer is played out when a time series of afferent spikes, rather than one synchronous discharge, generate a postsynaptic spiking response (*Thomson, 2000*; *Williams and Stuart, 2002*; *Williams and Atkinson, 2007*; *Grillo et al., 2018*). Therefore, in the second test, we simulated a Poisson process of stochastic synaptic discharges, across the physiological range of frequencies, and monitored the postsynaptic cell output. Experimental attempts to explore stochastic input to principal neurons have often employed synchronous (extracellular) activation of multiple afferent fibres, which is unlikely to happen in an intact brain where individual inputs can generate independent spike series. Thus, relatively intense synaptic input activity seen in pyramidal cells in vivo could correspond to a relatively low spiking frequency at the individual multiple afferents converging onto these cells (*Bähner et al., 2011*; *Kowalski et al., 2016*). These considerations suggest that short-term plasticity, if any, of intense synaptic input is actually driven mainly by the evolution of $P_r$ values over low or moderate spiking frequencies at individual axons.

To understand whether and how the distance dependence of $P_r$ influences spiking output of the postsynaptic pyramidal cell, we first compared the case of uniform $P_r$ values ($P_r = 0.36$), with that of the $P_r$ trend (the spine density trend was ignored as it had a negligible effect on EPSCs; *Figure 4B*). The outcome shows that having the $P_r$ trend provides a clear advantage for signal transfer fidelity, which is particularly prominent at higher input frequencies (*Figure 4C*). However, in these simulations we assumed no use-dependent changes in the transmission efficacy as set by $P_r$. In fact, a recent study of the CA3-CA1 circuitry has established the time-dependent degree of (presynaptic) short-term plasticity (STP) during afferent bursts, over a wide range of firing frequencies (*Mukunda and Narayanan, 2017*). In brief, during repetitive presynaptic activity, $P_r$ undergoes

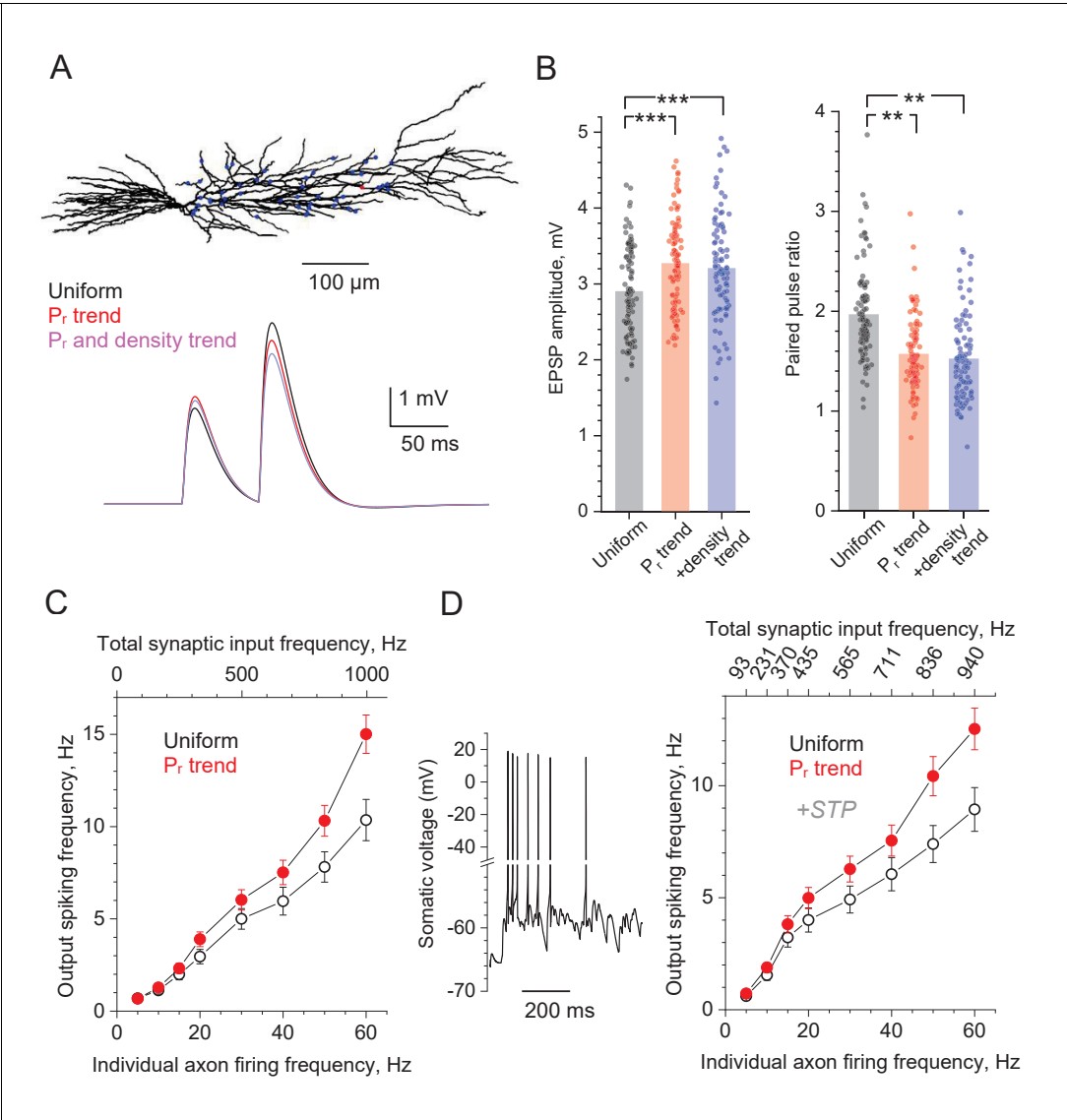

**Figure 4.** Computer simulations of CA1 pyramidal cell with stochastic excitatory synapses. (**A**) Diagram, NEURON model of a reconstructed CA1 pyramidal cell (*Migliore et al., 1999*) (ModelDB 2796; variable time step *dt*, t = 34˚C). 50 excitatory inputs (blue dots) generate bi-exponential conductance change (rise and decay time, 1 ms and 20 ms, respectively) stochastically, in accord with $P_r$. Traces, simulated somatic response to paired-pulse stimuli (50 ms apart), with $P_r$ distributed either uniformly randomly (black), or in accord with the distance-dependent trend (as in *Figure 1C*; red), and both $P_r$ and synaptic density trends (as in *Figure 2E*, blue; same average $P_r$ = 0.36 ). (**B**) Summary of simulation tests in (**A**); dots, individual runs (n = 100); bars, mean EPSP amplitude (left, mean ± SEM: 2.90 ± 0.060, 3.27 ± 0.061, 3.20 ± 0.068, for the three conditions, respectively, as indicated) and paired-pulse ratios (right, mean ± SEM: 1.96 ± 0.047, 1.57 ± 0.036, 1.52 ± 0.0437, notation as above) are shown; ***p<0.005. (**C**) Input-output spiking rate relationship over the physiological range of input firing frequencies (per axon, bottom axis; total, top axis); hollow circles, uniform distribution of $P_r$; solid symbols, $P_r$ follows the distance-dependent trend (as in *Figure 1C*); mean ± SEM are shown (n = 100 simulation runs). (**D**) Trace: A characteristic cell spiking burst (model as in A) in response to a Poisson-process afferent spiking input (~50 Hz per synapse) incorporating the experimental kinetics of short-term plasticity (STP) at CA3-CA1 synapses (*Mukunda and Narayanan, 2017*) (see *Figure 4—figure supplement 1C–F* for detail). Graph: Input-output spiking rate relationship across the physiological range of average input firing frequencies, with experimental STP incorporated; other notations as in (**C**); the top abscissa scale is nonlinear because STP affects average $P_r$ in a biphasic, non-monotonous manner (see *Figure 4—figure supplement 1C*).

The online version of this article includes the following source data and figure supplement(s) for figure 4:

**Source data 1.** Original data readout for *Figure 4A–D* and *Figure 4—figure supplement 1A–F*.

**Figure supplement 1.** Simulating the experiment-based kinetics of short-term plasticity (STP) in the CA3-CA1 circuit.

prominent facilitation and depression within the first 2–4 discharges, followed by a near-constant $P_r$ value that depends on the average input frequency (*Mukunda and Narayanan, 2017*). We have incorporated this STP algorithm (*Figure 4—figure supplement 1C*) and successfully validated it in our model by comparing simulated and recorded outcomes of the fixed-frequency afferent bursts (*Figure 4—figure supplement 1D–F*). Equipped with this STP mechanism, we generated Poisson-process afferent input in our model and found, once again, that the $P_r$ trend improves signal transfer, with the effect monotonically increasing with higher frequencies of afferent firing (*Figure 4D*).

## Discussion

The way information is handled and stored by brain circuits depends largely on the rules of synaptic signal integration by principal neurons. Passive electric properties of neuronal dendrites impose distance-dependent attenuation of synaptic receptor current. Thus, distal synapses should have a progressively weaker influence on somatic spike generation. If true, this would mean that neurons with large dendritic trees possess a highly inefficient synaptic connectome. However, studies in hippocampal CA1 pyramidal cells have found that synaptic receptor numbers increase with the synapse-soma distance (*Andrasfalvy and Magee, 2001*; *Nicholson et al., 2006*) and that ion channel properties at distal dendrites enable local regenerative events (*Cash and Yuste, 1999*; *Magee, 1999*). These features seem to underpin near-linear summation of synaptic inputs across the entire tree (*Cash and Yuste, 1999*; *Magee, 2000*; *Magee and Cook, 2000*), thus rescuing the efficiency of dendritic signal integration. However, equal contribution of individual axo-dendritic synapses to somatic signal assumes that their average discharge rates are similar. This in turn directly depends on $P_r$ and its use-dependent plasticity at individual synapses (*Williams and Stuart, 2002*; *Williams and Atkinson, 2007*). It was therefore a surprising discovery that the $P_r$ values at the basal dendrites of CA1 pyramidal cells showed a centrifugal decrease, at least in conditions of synchronous, multi-synaptic paired-pulse stimuli (*Grillo et al., 2018*). If common, this trend, again, would question the efficiency of synaptic connectome: it seems wasteful to make so many distal synaptic connections that have so little impact.

### Assessing $P_r$ using optical quantal analysis and paired-pulse ratios

Here, we used OQAs to measure $P_r$ at individual synapses in apical dendrites of CA1 pyramidal cells and found the opposite: $P_r$ values were increasing with the synapse-soma distance along the dendrites. There are certain advantages of the OQA over electrophysiological PPR measurements. Firstly, OQA requires only single-pulse stimulation and establishes absolute, rather than relative, $P_r$ values and their distribution pattern along the dendrites. Secondly, OQA deals with confirmed afferent activation of individual synapses and thus avoids the potential dependence between excitability of an axon and $P_r$ value at its synapses, which would be masked in multiple-fibre stimulation. Thirdly, intrinsic differences in the short-term plasticity kinetics among different synaptic populations may contribute to the variation in PPRs. As for the $P_r$ assessment based on the use-dependent NMDA receptor blocker MK-801, the progressive blockade of the NMDAR current may involve a degree of local inter-synaptic NMDAR activation via glutamate spillover (*Scimemi et al., 2004*). The latter should have a much smaller effect on the postsynaptic optical $Ca^{2+}$ signal which, classically, displays S-shape sensitivity to the underlying NMDAR current, so, it is virtually insensitive to small current increments near the bottom (release failure) and near the top (release success) of the scale. Indeed, multiple studies involving classical OQA or single-synapse LTP induction have indicated that, under suitable protocols, postsynaptic $Ca^{2+}$ signals elicited either by local afferent stimuli or by glutamate uncaging, do not spread to neighbouring spines (*Oertner et al., 2002*; *Emptage et al., 2003*; *Boddum et al., 2016*; *Henneberger et al., 2020*).

The technically strenuous OQA has its own limitations, of which the main one is the low signal-to-noise ratio, at least at some connections. In addition, the high-affinity (slowly unbinding) $Ca^{2+}$ indicators required to detect dendritic spine signals tend to saturate rapidly, thus preventing reliable $P_r$ readout for more than ~2 successive stimuli. We therefore, firstly, focused on synapses that showed a clear distinction between failures and successes (*Figure 1B*, *Figure 1—figure supplement 1C,D*), noting that this sampling still safely included the $P_r$ range from 0.1 to 0.9 (*Figure 1D*). Secondly, we used two PPR readouts as a gauge for $P_r$ variations, with the adjusted PPR measure avoiding any

uncertainty about second-pulse detection of successes or failures (*Figure 1E*). Both PPR measures pointed to a clear centrifugal trend for $P_r$ values (*Figure 1D–E*).

To further address a potential bias in selecting dendritic spines, we employed an alternative experimental design, in which $P_r$ trends were assessed in the bulk of synapses in area CA1 by imaging evoked glutamate release with the genetically encoded optical sensor iGluSnFR. Similar to the earlier method (*Grillo et al., 2018*), we used PPR values to assess position-dependent variations in $P_r$. While the earlier method referred to the position of the stimulation electrode with respect to the patched postsynaptic cell morphology, the present approach provided an exact spatial reference to the site of glutamate release in the *s. radiatum*. As explained in the Results, iGluSnFR signals registered here under extracellular stimuli are most likely to reflect release of glutamate from several local synapses. This makes such an approach suitable for PPR measurements rather than for direct $P_r$ estimates. The outcome of these experiments was consistent with the OQA data.

In the $P_r$-related data scatters presented here, Pearson's correlation r was relatively low (*Figures 1–2*). Indeed, the unexplained variance in the scatters, which is $(1-r^2)$, reflects large biological variability of $P_r$ among synapses, as expected. Classically, low Pearson's r between variables *x* and *y* is interpreted as 'weak' correlation, because in such cases sampling a certain *x* value provides poor prediction for the corresponding *y* value, and vice versa. This prediction power could be critical, for instance, in a clinical setting where measurement *x* from an individual is used to gauge where this individual is on the *y* scale. However, in establishing the significance of a population-average trend such prediction power is often irrelevant because the uncertainty about the trend will diminish with sampling (which would effectively cancel out the symmetrical noise around the trend). For instance, pooling and averaging $P_r$ values over 25 µm distance bins in *Figure 1C* data will increase Pearson's r to 0.80 although this would represent the same $P_r$ dataset. In the pooling exercise, we could sample 'groups of synapses' rather than individual synapses because multi-synapse recording is allowed by the context, as in *Figure 3* data or in earlier studies (*Grillo et al., 2018*). Thus, we must consider the statistical significance of the trend (p value for the non-zero slope) as a factor of prevailing importance, in the present context.

## Possible implications for dendritic signal integration

We have thus found that both classical OQA and the iGluSnFR approach point to the centrifugal increase of $P_r$ in apical dendrites of CA1 pyramidal cells. Intriguingly, an earlier study has found that at CA3-CA1 synapses release probability detected with OQA varies with the size of synaptic contacts (*Holderith et al., 2012*). This finding, combined with the evidence for greater AMPAR receptor numbers at more distal synapses (*Andrasfalvy and Magee, 2001*; *Nicholson et al., 2006*), is fully consistent with the notion of $P_r$ increasing with the distance to the soma in these cells. Such data lend support to the idea that synaptic organisation can powerfully compensate for the 'electrotonic weakness' of synapses in distal dendrites. The effects found here are similar to the location-dependent features of synaptic inputs revealed with paired recordings between L2/3 and L5 pyramidal neurons in the neocortex (*Williams and Stuart, 2002*; *Williams and Atkinson, 2007*). Clearly, the occurrence of the higher-efficacy synapses at distal dendrites must boost temporal summation of local CA3-CA1 inputs (*Stuart and Spruston, 2015*). This is consistent with the observation that LTP induction there involves local sodium spikes (*Kim et al., 2015*). Furthermore, it has recently been found that individual CA3-CA1 axons make multiple synapses in distal (rather than proximal) dendrites of CA1 pyramidal cells (*Bloss et al., 2018*). This further argues for added efficiency of distal synaptic connections, and is in line with the relatively high dendritic spine density found here. Because the spine density estimate could be affected by limited resolution of two-photon excitation imaging (which may not necessarily detect smallest or thinnest spines), it would be prudent to assume that here we refer mainly to larger, mushroom-type dendritic spines.

Notwithstanding methodological differences, comparing the present results with the earlier findings (*Grillo et al., 2018*) suggests that the dendritic integration traits at basal and apical inputs in CA1 pyramidal cells are starkly different. Excitatory inputs to basal dendrites in the *s. oriens* include axons of other pyramidal cells, septal fibre, and commissural axons from the contralateral hippocampus. Interestingly, the spine density here increases almost twofold away from the soma (*Bannister and Larkman, 1995*; *Ballesteros-Yáñez et al., 2006*). Whether and how these different inputs cluster within different parts of the basal dendrites, and how their spiking activities differ, might shed light on the adaptive role of the apparently 'inefficient' $P_r$ pattern there. For instance,

biophysical modelling of a simplified neuron suggested that the $P_r$ and PPR trends in basal dendrites help to facilitate supra-linear summation of local generative events such as dendritic spikes (*Grillo et al., 2018*). Indeed, such events have long been considered as a key mechanism of distal dendritic signalling (*Larkum and Nevian, 2008*; *Branco and Häusser, 2009*).

In the present study, we explored a realistic model of a CA1 pyramidal cell (*Migliore et al., 1999*) hosting excitatory synapses that incorporated varied $P_r$ values, and were scattered across the apical dendrites, in accord with our experimentally observed trends. Simulations suggested that having the centrifugal $P_r$ trend boosts synaptic signal transfer across the physiological frequency range, with the effect monotonically increasing with higher firing frequencies. Importantly, in an intact rat brain, CA1 pyramidal cells tend to fire in short, high-frequency bursts (*Bähner et al., 2011*; *Kowalski et al., 2016*) associated with sharp-wave ripple complexes (*Bähner et al., 2011*). The latter type of synchronised neuronal activity is thought to represent a powerful mechanism of memory consolidation in the hippocampus (*Gruart et al., 2006*; *Buzsáki, 2015*). In fact, short bursts of activity have long been considered as a key unit of information transfer by stochastic synapses (*Lisman, 1997*). Thus, the $P_r$ trend found here in apical dendrites must play an essential role in supporting and enhancing the cellular underpinning of learning and memory in the brain.

# Materials and methods

## Key resources table

| Reagent type (species) or resource | Designation | Source or reference | Identifiers | Additional information |
|---|---|---|---|---|
| Chemical compound, drug | Picrotoxin | Tocris | Cat. # 1128 | |
| Chemical compound, drug | D-Serine | Sigma Aldrich | Cat. # S4250 | |
| Chemical compound, drug | Oregon Green 488 BAPTA-1, Hexapotassium Salt | ThermoFisher | Cat. #O6806 | |
| Chemical compound, drug | Alexa Fluor-594 Hydrazide | ThermoFisher Scientific | Cat.# A10438 | |
| Chemical compound, drug | Fluo-8, potassium salt | Stratech Scientific | Cat.# 21087-AAT | Original Source: (Teflabs) No longer exists |
| Chemical compound, drug | Fluo-4, Pentapotassium Salt | ThermoFisher Scientific | Cat.# F14200 | |
| Chemical compound, drug | Agarose | Sigma-Aldrich | Cat. #9539; CAS: 9012-36-6 | Freshly prepared (4%) |
| Chemical compound, drug | D-glucose | Sigma-Aldrich | Cat. #G8270; CAS: 50-99-7 | Freshly prepared (10 mM) |
| Chemical compound, drug | KCl | Sigma-Aldrich | Cat. #P9333; CAS: 7447-40-7 | |
| Recombinant DNA reagent | AAV9.hSynap. iGluSnFr.WPRE. SV40 | Penn Vector Core | Addgene Cat. # 98929-AAV9 | |
| Strain, strain background (Sprague-Dawley rat) | Sprague-Dawley | Charles River UK | Crl: CD (SD) Strain: 0204 | |
| Strain, strain background (C57BL/6J mouse) | C57BL/6J | Charles River UK | C57BL/6NCrl Strain: 0159 | |

*Continued on next page*

*Continued*

| Reagent type (species) or resource | Designation | Source or reference | Identifiers | Additional information |
|---|---|---|---|---|
| Software, algorithm | ImageJ | NIH | RRID:SCR_003070 https://imagej.nih.gov/ij/ | |
| Software, algorithm | pClamp10 | Molecular Devices | RRID:SCR_011323 https://www.moleculardevices.com/products/axon-patch-clamp-system/acquisition-and-analysis-software/pclamp-software-suite | |
| Software, algorithm | OriginPro | OriginLab Inc | RRID:SCR_014212 https://www.originlab.com/origin | |
| Software, algorithm | MES 4.x-5.x | Femtonics Ltd. | RRID:SCR_018309 https://uk.mathworks.com/products/connections/product_detail/femtonics-mes.html | |
| Software, algorithm | WinWCP Versions 4.x-5.x | Strathclyde Electrophysiology Software | RRID:SCR_014270 http://spider.science.strath.ac.uk/sipbs/software_ses.htm | |
| Software, algorithm | Matlab | Mathworks | RRID:SCR_001622 https://uk.mathworks.com/products/matlab.html | |
| Software, algorithm | NEURON 7.6 × 64 | https://www.neuron.yale.edu/neuron/ | RRID:SCR_005393 | |
| Software, algorithm | Kinetics of CA1 pyramidal neuron | https://senselab.med.yale.edu/ModelDB/ShowModel?model=2796#tabs-2 | RRID:SCR_005393 model = 2796#tabs-2 | |
| Software, algorithm | reconstructed CA1 pyramidal neurons | https://senselab.med.yale.edu/ModelDB/ShowModel?model=7509#tabs-1 | RRID:SCR_005393 model = 7509#tabs-1 | |
| Other | Multiclamp 700B | Molecular Devices | RRID:SCR_018455 | |
| Other | Olympus FluoView1000 | Olympus | RRID:SCR_014215 | |
| Other | Femto3D RC | Femtonics | Femto3D RC | |
| Other | BioRad Radiance 2100 | BioRad | Radiance 2100 | |
| Other | Leica VT1200S vibratome | Leica Biosystems | RRID:SCR_018453 | |

## Animals and hippocampal slice preparation

Animal procedures were subject to local ethical approval and adhered to the European Commission Directive (86/609/ EEC) and the United Kingdom Home Office (Scientific Procedures) Act of 1986. Male Sprague Dawley rats (P20-P28) were sacrificed using an overdose of isoflurane. Animals were kept in groups (5–8 per cage) under standard housing conditions with 12 hr light–dark cycle and free access to food pellets and drinking water. Following decapitation brains were rapidly removed, hippocampi dissected out and transverse slices (350 µm thick) were prepared on a Leica VT1200S vibratome. The slicing was performed in ice-cold slicing solution that contained (in mM): 60 NaCl, 105 sucrose, 26 $NaHCO_3$, 2.5 KCl, 1.25 $NaH_2PO_4$, 7 $MgCl_2$, 0.5 $CaCl_2$, 11 glucose, 1.3 ascorbic acid, and three sodium pyruvate. Alternatively, slices were cut in an ice-cold slicing solution that contained (in mM): 64 NaCl, 2.5 KCl, 1.25 $NaH_2PO_4$, 0.5 $CaCl_2$, 7 $MgCl_2$, 25 $NaHCO_3$, 10 D-glucose, and 120 sucrose. All solutions were bubbled with 95% $O_2$ plus 5% $CO_2$, pH adjusted to 7.4. Once cut, slices were incubated in the oxygenated slicing solution at 34°C for 15 min and then allowed to equilibrate to room temperature for 15 min following which they were transferred to either a continuously oxygenated humid interface chamber, or a submersion chamber. Both chamber types contained an oxygenated artificial cerebrospinal fluid (aCSF) solution containing (in mM): 120 NaCl, 10 glucose, 2.5

KCl, 1.3 $MgSO_4$, 1 $NaH_2PO_4$, 25 $NaHCO_3$, 1.3 $MgCl_2$, 2 $CaCl_2$, with an osmolality of ~300 mOsm. Slices were then allowed to recover at room temperature for 1–5 hr prior to being transferred to the recording chamber of the microscope constantly perfused with 32–34°C.

## Electrophysiology ex vivo

Whole-cell patch-clamp recordings were performed from CA1 pyramidal cells visualised using infrared differential contrast imaging. Thin-walled borosilicate glass capillaries were used to fabricate recording electrodes with a resistance of 2.5–3.5 MΩ. Intracellular pipette solution contained (in mM) $KCH_3O_3S$ 135, HEPES 10, $Na_2$-Phosphocreatine or di-Tris-Phosphocreatine 10, $MgCl_2$ 4, $Na_2$-ATP 4, Na-GTP 0.4, 5 QX-314-Bromide (pH adjusted to 7.2 using KOH, osmolarity 290–295). Cell-impermeable $Ca^{2+}$ dyes detailed below and the $Ca^{2+}$ insensitive morphological tracer Alexa Fluor 594 hydrazide (50 µM) were routinely added to the intracellular solution. Throughout recordings cells were held in voltage clamp at −65 mV, and the aCSF routinely supplemented with 100 µM Picrotoxin (Tocris Bioscience) and 30 µM D-Serine. Electrophysiological recordings were carried out by using a set of remotely controlled micromanipulators and XY-translation stage (Luigs and Neumann) with a Multiclamp 700B amplifier (Molecular Devices). Signals were digitised at 10 kHz and stored for off-line analysis using WinWCP V4.1.5–4.7 (John Dempster, University of Strathcyde) or PClamp 10.4–10.5 (Molecular Devices).

## Two-photon excitation imaging: optical quantal analysis

We used a Radiance 2100 imaging system (Zeiss–Bio-Rad; 60x Olympus objective, NA0.9) or a Femtonics Femto3D-RC imaging system (25x Olympus objective, NA1.05) both optically linked to two femtosecond pulse lasers MaiTai SpectraPhysics-Newport and integrated with patch-clamp electrophysiology.

Intracellular $Ca^{2+}$ responses were routinely imaged in the $Ca^{2+}$-sensitive (green) emission channel filtered to the bandwidth of 500–550 nm, with either of the $Ca^{2+}$ sensitive dyes Oregon Green BAPTA-1 (250 µM, Invitrogen), Fluo-4 (400 µM, Invitrogen) or Fluo-8 (300 µM, Stratech Scientific). For optical quantal analysis, synaptically evoked responses in dendritic spines were triggered by minimal electrical stimulation using a monopolar glass stimulating electrode filled with aCSF and placed 10–20 µm from the dendritic branch targeted. To identify active synapses relatively fast (10 Hz) frame scans of the local dendrites were viewed whilst three 100 µs square pulses of 2–10 V were delivered with a 25 ms inter-stimulus interval using a constant voltage isolated stimulator (model DS2A-mkII, Digitimer Ltd, Welwyn Garden City, UK). This protocol was repeated until a $Ca^{2+}$ response confined to a spine head was observed, then 400–700 ms line scans of the active spine were then recorded at a line scan rate of 500 Hz. In order to reduce the inherent bias toward high release probability synapses, a dual stimulus protocol (50 ms inter-stimulus interval) was employed to ensure response stability at synapses with low initial release probability. Line scans were repeated once every 30 s with 14-26 trials to assess release probability while correcting manually for focus fluctuations in the Alexa (red) channel. Line scan profiles were extracted using ImageJ 1.x (*Schneider et al., 2012*) or Femtonics MES six and routinely documented as $\Delta G/R$ where $\Delta G = G - G_0$ stands for the fluorescence signal in the green channel $G$ with the baseline fluorescence $G_0$ (averaged over the baseline time window) subtracted, and $R$ stands for the Alexa fluorescence in the red channel (corrected for photobleaching, if any). Where recordings of synaptic responses were made at two sites along a single dendritic branch (distal and proximal to the branch point), equal numbers of recordings, first proximal–distal and distal–proximal, were obtained to remove bias in the temporal order by which the recordings were carried out. Experiments were excluded if the synapse became non-responsive, if spine morphology visibly changed during recording or if significant drift of the stimulation electrode occurred. Release probability ($P_r$) was defined as the probability of success on the first stimulus, as shown as P1 in figures: sum of successful trials / total number of trials, or {(1,0) + (1,1)} / (all); here brackets indicate counts of successes (1), failures (0), and all paired-stimulus trials (all). Where a second response was clearly visible over the first, P2 was also measured and defined as the sum of responses to the second stimulus + responses to both stimuli / total number of trials, or {(0,1) + (1,1)} / (all). PPR was defined as P2/P1 = P2/$P_r$. To account for some uncertainty over the identification of (1,1) trials (because of $Ca^{2+}$ signal saturation upon the second stimulus), we

also calculated adjusted PPR* = P2*/P1 where P2* = (0,1) / {(0,0) + (0,1)}, so that (1,1) counts were ignored.

## Two-photon excitation imaging: morphological measurements

For morphological tracing purposes, wide-field high-resolution images of the apical dendritic tree (>75 µm$^2$) were acquired at the maximal optical resolution as a series of individual 3-D stacks (50–100 µm deep each) in the Alexa emission channel (550–650 bandwidth; $\lambda_x^{2P}$ = 800 nm), collected in image frame mode (Biorad: 512 × 512 pixels, 8-bit; Femto3D > 512×512 pixels, 16-bit) at 1.5–2.5 µm steps. These images were used to determine the location of the individual synapses in relation to the CA1 pyramidal cell soma, first the averaged Alexa 594 image stack was used to form a template defining the two-dimensional x-y path (along the dendrite) between soma and spine of interest. The start point of the distance measurement was defined as the centre of an oval shaped to fit the edges of the imaged soma, from this point the segmented line function in ImageJ was used to trace the path along the dendritic tree between the soma and the spine of interest ending at the base of the spines shaft. This line was then saved as a region of interest and then overlaid onto the original 3D image stack. The 'reslice' function in ImageJ was then used to form a two-dimensional y-z image in which the y-axis represents the length of the template line and the z-axis representing the deviation in the depth of the slice thus enabling simple measurement of distance from soma to synapse also accounting for z-plane deviation. With this method, the distances from the origin of and to the end of the parent dendrite were also obtained. Higher magnification images of the same resolution were also acquired and the density of spines local to the synapse estimated by counting the total number of visible protrusions along a 10 µm portion of dendrite centred on the synapse of interest.

## Viral transduction for labelling Schaffer collateral axonal boutons

For ex vivo imaging of axonal boutons, viral transduction in vivo via neonatal intracerebroventricular (ICV) injections in both male and female C57BL/6J mice (Charles River Laboratories) were used, as we detailed earlier (*Henneberger et al., 2020*; *Kopach et al., 2020*). Briefly, an AAV virus expressing the neuronal optical glutamate sensor, AAV9.hSynap.iGluSnFR.WPRE.SV40 (supplied by Penn Vector Core, PA) was injected into the cerebral ventricles of neonates (P0-P1) during aseptic surgery. The viral particles were injected at 2 µl per hemisphere (2.5–5 × 10$^9$ genomic copies in total), at a rate not exceeding of 0.2 µl/s, 2 mm deep, guided to a location approximately 250 µm lateral to the sagittal suture and 500–750 µm rostral to the neonatal coronary suture. After animals received AAV injections, they were returned to the mother in their home cage; they were systematically kept as a group of litters and monitored for days thereafter, to ensure that no detrimental side effects appear. Satisfactory transduction of iGluSnFR in vivo occurred within 3–4 weeks.

Two-photon excitation (2PE) imaging of glutamate release with iGluSnFR iGluSnFR fluorescence was recorded in the green emission channel under 2PE at $\lambda_x^{2P}$=910 nm, using a Femtonics Femto3D-RC imaging system or an Olympus FV10MP imaging system, both optically linked to a Ti:Sapphire MaiTai femtosecond-pulse laser (SpectraPhysics-Newport), equipped with galvo scanners, and integrated with patch-clamp electrophysiology. In the *s.radiatum,* we focused on individual axonal boutons that could be visualised by iGluSnFR expression and showed a consistent optical response to electric stimuli applied to Schaffer collaterals. To minimise photodamage, only a single focal section through the region of interest (ROI) containing selected axonal fragments was imaged, at laser power not exceeding 3–6 mW under the objective. The focal plane was regularly adjusted, to account for specimen drift. To record optical signals with high spatiotemporal resolution while minimising photodamage, we employed the scanning mode of spiral ('tornado') linescans centred at the bouton of interest, as detailed previously (*Jensen et al., 2017*; *Jensen et al., 2019*). In these experiments, we recorded responses to paired-pulse stimuli at 20 Hz, normally collecting 20–30 trials ~ 30 s apart, and documented the shortest distance between the recorded bouton and the *s. pyramidale* border. The iGluSnFR fluorescence response to afferent stimulation was expressed as $(F(t)- F_0) / F_0 = \Delta F / F_0$, where $F(t)$ stands for fluorescence intensity over time, and $F_0$ is the baseline intensity averaged over ~150 ms prior to the first stimulus.

In a complementing experiment, we applied five-pulse stimuli (at 20 Hz) to Schaffer collaterals and used time-lapse imaging in frame-scan mode, as detailed previously (*Kopach et al., 2020*), thus providing short pixel dwell time (~0.5 µs) to enable signal registration over larger imaged regions,

albeit at the expense of temporal resolution. This approach was chosen because regions of interest had to be selected off-line post hoc, based on their overall morphological stability: burst stimulation could lead to microscopic focal drifts or tissue changes in some areas, which had to be avoided. Therefore, the experimental time course of iGluSnFR featuring a limited number of time points was, in some cases, fitted with the underlying theoretical kinetics of five overlapping $\Delta F / F_0$ signals so that: $\Delta F/F_0 = \sum A_i exp(-(t - \Delta t \cdot (i-1)) \cdot \tau^{-1})$ ($i = 1,..., 5$) where $A_i$ is the $i$th signal amplitude (fitted directly to the recoded amplitude), $\Delta t$ = 50 ms (inter-spike interval), and the decay constant $\tau$ obtained from fitting the signal decay (tail) after the fifth pulse.

## NEURON modelling

We employed a multi-compartmental NEURON (*Hines and Carnevale, 2001*) model of a reconstructed CA1 pyramidal neuron, with a full set of experiment-adjusted membrane currents, uploaded from the NEURON 7.6 × 64 database (https://senselab.med.yale.edu/modeldb; models 2796 and 7509). Simulations were performed using a variable time step $dt$; the cell axial-specific resistance $R_a$ and capacitance $C_m$ were set at $R_a$ = 90 Om·cm, and $C_m$ = 1 mF/cm$^2$, throughout simulations. Fifty synapses were distributed over the dendritic apical tree, between 40 and 350 μm from the soma, either uniformly randomly or scattered randomly in accord with the experimental statistical trend, as specified. Computation tests for any set of parameters routinely consisted of 100 runs (trials). To avoid any bias arising from a particular (fixed) distribution of synapses along the dendrites, this distribution was generated anew at each trial, albeit with the same probability density function. However, in the tests where non-uniform and uniform distributions of $P_r$ were compared, the random generator seed was kept unchanged.

At individual synapses, excitatory synaptic conductance $g_s(t)$ was modelled using the dual-exponential formalism realised in the NEURON function Exp2Syn. The dynamics models synaptic activation as a change in synaptic conductance with a time course given by: $g_s = G_m(exp(-t/\tau_1) - exp(-t/\tau_2))$ where $\tau_1$ and $\tau_2$ are the rise and decay time constants, respectively, and $G_m$ is the factor defined by the peak synaptic conductance. The reverse potential of excitatory synapses was zero, in line with that for AMPA and NMDA receptor types. In line with the data reporting that synaptic strength increases centrifugally in apical dendrites of CA1 pyramidal cells (*Andrasfalvy and Magee, 2001*; *Nicholson et al., 2006*), $G_m$ values were set proportional to the function $f(x)$ = $A$·(0.51 + 0.002$x$) where $x$ is the synapse-soma distance, and $A$ is scaling factor (NEURON NetCon object), which in our case sets single-synapse conductance $G_m$ (in μS). In simulations exploring subthreshold postsynaptic excitation, $A$ was set at 0.002 μS, and in tests exploring the spiking input - output relationship $A$ was set at 0.06 μS.

To replicate our experimental observations of release probability, the average $P_r$ (P1) was set at 0.36, and P2 at 0.77. These values remained constant throughout the synaptic population in the case of the uniform $P_r$ distribution. In the case of the experimental (centrifugal) $P_r$ trend, the $P_r$ (P1) and P2 values were set as a function of the synapse-soma distance $x$: P1 = (0.18 + 0.0012$x$), and P2 = (0.18 + 0.0012$x$) · (3.31–0.0074$x$), reflecting experimental regressions, with the average $P_r$ value kept at 0.36 throughout, to ensure the unchanged overall efficacy of integrated synaptic input. Calculations started 300 ms following full stabilisation of the NEURON model.

## Sampling and statistics

The present study contained no longitudinal experiments or biological hypotheses based on multiple comparisons or experimental samples involving bootstrapping. Dendritic spines or axonal boutons were sampled in an arbitrary manner, as they appeared in the focal plane showing distinct morphology and clear (above two standard deviations of the no-response noise) stimulus-induced fluorescence responses. No exclusion criteria were applied to animals or slices, experiment sessions were discarded if the dendritic spine or axonal bouton became non-responsive, or if a significant drift of the stimulation electrode or preparation occurred. Blinding was not applicable to experimental manipulations during live recording sessions.

In real-time experiments, the statistical units were either individual dendritic spines, individual axonal boutons, or groups of axonal boutons where applicable, with the effect of experimental manipulation being the only factor of interest. In OQA experiments, 1–2 dendritic spines were successfully recorded from individual CA1 pyramidal cells, 1–2 cells were recorded per slice / animal. In

iGluSnFR experiments, axonal boutons or group of boutons were sampled as explained in the text. The paired 'baseline-effect' comparison was therefore employed in all such experiments, in accord with the convention of electrophysiological experiments. The sample size was not predetermined because the variability of measured parameters - be it the variance for mean estimates or the unexplained Pearson's variance for regression estimates - was not known a priori, and there was no known restriction or bias in the cohort sampling. No repeated measures were included in testing biological hypotheses. Experimental readout involved live recording sessions carried out once per statistical unit.

Statistical data were routinely presented as mean ± SEM and sample size. Testing statistical significance of correlation / linear regression, involved a standard *t*-test for the slope parameter (equivalent to the correlation test), and included Pearson's r. Shapiro-Wilks tests for normality produced varied results across raw data samples. Accordingly, we used either the paired-sample *t*-test, or the paired-sample non-parametric Wilcoxon Signed Ranks test, where indicated. The statistical software in use was Origin 2019 (Origin Lab; RRID: SCR_014212).

## Acknowledgements

This study was supported by the Wellcome Trust Principal Fellowship (212251_Z_18_Z), ERC Advanced Grant (323113) and European Commission NEUROTWIN grant (857562).

## Additional information

### Funding

| Funder | Grant reference number | Author |
| --- | --- | --- |
| Wellcome Trust | 212251_Z_18_Z | Dmitri A Rusakov |
| European Research Council | 323113 | Dmitri A Rusakov |
| European Commission | 857562 | Dmitri A Rusakov |

The funders had no role in study design, data collection and interpretation, or the decision to submit the work for publication.

### Author contributions

Thomas P Jensen, Conceptualization, Formal analysis, Investigation, Visualization, Methodology, Writing - review and editing; Olga Kopach, Formal analysis, Investigation, Visualization, Methodology, Writing - review and editing; James P Reynolds, Methodology; Leonid P Savtchenko, Software, Formal analysis, Investigation, Methodology, Writing - review and editing; Dmitri A Rusakov, Conceptualization, Resources, Data curation, Formal analysis, Supervision, Funding acquisition, Writing - original draft, Project administration, Writing - review and editing

### Author ORCIDs

Olga Kopach ⓘD http://orcid.org/0000-0002-3921-3674
Dmitri A Rusakov ⓘD https://orcid.org/0000-0001-9539-9947

### Ethics

Animal experimentation: Animal procedures were subject to local ethical approval and adhered to the European Commission Directive (86/609/ EEC) and the United Kingdom Home Office (Scientific Procedures) Act of 1986. Experiments were carried out under the UK HO Project licence PPL707524.

### Decision letter and Author response

Decision letter https://doi.org/10.7554/eLife.62588.sa1
Author response https://doi.org/10.7554/eLife.62588.sa2

## Additional files

### Supplementary files

• Transparent reporting form

### Data availability

All data generated or analysed during this study are included in the manuscript and supporting files. Source data files have been provided for Figures 1–4.

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
