## [Decision Letter]

Thank you for submitting your article "Release probability increases towards distal dendrites boosting high-frequency signal transfer" for consideration by *eLife*. Your article has been reviewed by three peer reviewers, including Sacha B Nelson as the Reviewing Editor and Reviewer #1, and the evaluation has been overseen by John Huguenard as the Senior Editor.

The reviewers have discussed the reviews with one another and the Reviewing Editor has drafted this decision to help you prepare a revised submission.

As the editors have judged that your manuscript is of interest, but as described below that additional analyses are required before it is published, we would like to draw your attention to changes in our revision policy that we have made in response to COVID-19 (https://elifesciences.org/articles/57162). First, because many researchers have temporarily lost access to the labs, we will give authors as much time as they need to submit revised manuscripts. We are also offering, if you choose, to post the manuscript to bioRxiv (if it is not already there) along with this decision letter and a formal designation that the manuscript is "in revision at *eLife*". Please let us know if you would like to pursue this option. (If your work is more suitable for medRxiv, you will need to post the preprint yourself, as the mechanisms for us to do so are still in development.)

Summary:

Prior work has established that neuronal sensitivity to released transmitter scales with distance from the axon so as to mitigate limitations of integrating more distal synapses. Here, the authors show for the first time that presynaptic release of glutamate is also "scaled" with distance. This is an important finding that potentially changes how we think about synaptic integration in these well studied hippocampal neurons, but reviewers raise concerns over aspects of the analyses that need to be addressed. Acceptability of the manuscript will be assessed after it is clearer whether or not the reanalysis supports the central claims of the paper.

Title: It is *eLife* policy that the title should make clear the biological system under investigation. This could be accomplished by including reference to "…the mammalian hippocampus" or at least "hippocampus" or "hippocampal neurons."

Essential revisions:

The full reviews are included below to help clarify the concerns. Each of the concerns raised should be addressed, however many of these require simply textual clarifications. Key additional analyses required are:

1) The reanalysis suggested by reviewer #3.

2) An analysis of baseline noise distributions and of how apparent Ca+ and iGluSnFR "failure" signals vary with the site and intensity of stimulation, coupled to a more detailed quantitative argument as to why spillover affects the latter, but not the former measurements.

Reviewer #1:

The authors use optical quantal analysis and iGluSnFR measurements of glutamate release at CA3-CA1 synapses to convincingly demonstrate a gradient in release probability that favors distal synapses, matching in many respects, the distance-dependent scaling demonstrated postsynaptically by Magee and others. This is an elegant study that reveals a new feature of hippocampal synaptic physiology. Although the net effect is modest (~15%) the authors also use a realistic biophysical model to demonstrate that the observed gradient improves information transfer to CA1 neurons.

Figure 1: why are 1/3 of the synapses in C. missing in D. Were these synapses for which a second response was not clearly visible? Is this presumed due to saturation? Some brief description of the reason for exclusion should be provided.

Reviewer #2:

In this study, Jensen and colleagues use optical imaging of synaptic calcium signals and glutamate release and report that presynaptic release probability increases in CA1 pyramidal cell dendrites where more distal synapses have a higher release probability. These results complement earlier observations on the increase in postsynaptic efficacy in distal dendrites. While the findings presented in this manuscript address a fundamental question in synapse biology and are potentially important, further documentation is needed to ensure validity of these results and realize their full impact.

1) The findings presented here contradict an earlier thorough study by Magee and colleagues that addressed the same question and reached opposite conclusions. I think the authors should discuss this work (Smith et al., 2003) and address the potential discrepancy.

2) It is important to document baseline noise distributions for calcium indicator as well as glutamate probe measurements. It is critical to demonstrate the reliability of success/failure classification in individual recordings. What is the variation in response amplitudes? How do they compare to baseline fluorescence fluctuations in the same trace?

3) As the senior author and colleagues proposed 14 years ago PPR can be release-independent (Volynski et al., 2006) and therefore makes a poor predictor of release probability. Given the authors have near single synapse resolution, it is surprising that they heavily rely on PPR measure for their arguments. While paired pulse stimulation is a good tool to reduce the bias towards potential high Pr synapses, failure analysis (first response success probability) should be more emphasized over paired pulse ratios. It is worrisome that iGluSnFR experiments solely rely on PPR measure. The authors indicate that they cannot detect failures due to spill over contribution to their measurements. Wouldn't spill-over be more of a concern for calcium signals as they rely on NMDA receptors? Given this group's track record in the field, I would have expected more insight into this issue. I think the authors should at least make sure that this is not a simple technical issue (e.g. location and intensity of stimulation etc.).

Reviewer #3:

I am not able to assess the manuscript in its current form, as I have doubts about the extraction of quantal parameters from the fluorescence traces. The description in the methods does not seem to match the examples provided. I suggest re-analysis of the data (not just of the example traces!).

In the Materials and methods, P2 is defined as "the sum of responses to the second stimulus + responses to both stimuli/total number of trials". There are 4 possible outcomes of paired pulse stimulation: double failure (0,0), single responses (1,0 and 0,1) and double responses (1,1). The author's definition sounds like P2 = [(0,1) + (1,1)] / [(0,0) + (1,0) + (0,1) + (1,1)], which makes sense.

Looking at the example traces shown in Figure 1B and Figure 1—figure supplement 1C, however, the given values of P2 correspond to [(1,0) + (0,1) + (1,1)] / [(0,0) + (1,0) + (0,1) + (1,1)], which is actually a different quantity, (P1+P2). The resulting calculated PPRs are thus not correct.

Given the SNR of the (best) examples, it seems very risky to try to distinguish between (1,0) and (1,1). My recommendation would be to define P2 = (0,1) / [(0,0) + (0,1)], which can be reliably extracted. Unfortunately, this requires complete re-analysis of all raw data and may change results and conclusions.

[Editors' note: further revisions were suggested prior to acceptance, as described below.]

Thank you for submitting your article "Release probability increases towards distal dendrites boosting high-frequency signal transfer in the rodent hippocampus" for consideration by *eLife*. Your article has been reviewed by three peer reviewers, including Sacha Nelson as the Reviewing Editor and Reviewer #1, and the evaluation has been overseen by John Huguenard as the Senior Editor.

The reviewers have discussed the reviews with one another and the Reviewing Editor has drafted this decision to help you prepare a revised submission.

The authors perform sophisticated optical measurements of spine calcium and extracellular glutamate in hippocampal tissue to investigate synaptic properties on the long apical dendrite of CA pyramidal cells. They report that synapses close to the soma displayed pronounced paired-pulse facilitation and lower release probabilities compared to distal synapses. The correlations of these properties with synaptic location are statistically significant, but very weak, indicating that the major causes of variability are still not understood. Interesting for researchers in the field, exactly the opposite gradient of synaptic properties was recently reported for the basal dendrites of the same cell type.

Summary:

After receiving clarifications of the analysis methods in the initial round of revision, reviewer 3 has raised some additional concerns. Other points raised by reviewers 1 and 2 have been adequately addressed. Specific revisions are suggested below. The full text of review #3 is also included for clarity.

Essential revisions:

1) Reviewer 3 correctly points out that the effect size for the central observation is small. This does not necessarily preclude publication, but this point should be made clearer to the reader both by including the correlation coefficients in the figures and adding consideration of this issue to the Discussion.

2) The analysis of the burst experiments should be revised along the lines suggested below, or these experiments should be removed or the concern raised adequately rebutted.

3) Whether or not the average Pr and facilitation were kept constant for the different modeling scenarios should be clarified. If this is not the case, it will be necessary to revise the modeling portion of the study, or to make a convincing argument as to why this is not required in order to demonstrate the functional significance of the observed trend.

Reviewer #3:

Jensen et al., use calcium- and glutamate-imaging to assess the properties of presynaptic terminals impinging on the dendrites of CA1 pyramidal cells at different distances from the soma. They find a trend towards higher release probabilities and less facilitation at distal synapses. Synapses that are close together (< 40 um) had similar release probabilities. It is reassuring to see a similar distance-dependent decrease of paired-pulse ratios using two very different parameters, single spine calcium and local glutamate (pooled from several synapses). Opposite gradients have been described for synapses on the proximal dendrites of CA1 neurons (Grillo, 2018), which makes this study interesting and controversial. The authors then use a Neuron model to compare synaptic inputs with uniform properties to inputs that reflect the detected trend to higher Pr at distal inputs. Higher frequencies are indeed better transmitted in the model with a gradient in Pr. This seems to be a good way to test the intuition about the impact of the scaling effect, but I have some questions about the simulated scenarios (below).

While the spine calcium imaging experiments are performed with a local stimulation electrode positioned close to each responding spine, the glutamate experiments use a single stimulation electrode position in stratum radiatum, but read-out in different layers. Although GluSnFR expression is punctate, the optical measurements represent bulk extracellular glutamate and not individual synapses. Therefore, the distance from the simulation electrode is likely the key parameter determining max df/f at different positions. The paired pulse ratio is the only parameter that can be extracted from the glu imaging experiments.

1) The discussion of correlations is focused on their p-values (which indicate that the slope of a linear regression is not zero). Biologically more interesting are the correlation coefficients, which indicate (after squaring) that only a small part of the total variability in Pr is determined by synapse position. Correlations with R squared = 0.18 (Figure 1C) or 0.14 (Figure 2A) are considered "very weak", which is not necessarily what you expect given the title of the manuscript. As all data points are shown, readers will form their own opinion and realize that synapse position has only a weak impact on presynaptic properties. Still, I recommend showing the numerical value of r (or r squared) on all panels with linear fits (and relegate "n" to the figure legends).

2) Figure 3C shows strong facilitation at proximal synapses, linear behavior at distal synapses, which the authors take as supporting evidence that Pr is higher at distal synapses. However, the first pulse amplitude is similar (Figure 3D, averaging failures and successes), which the authors explain away by saying these measurements average glutamate from at least 60 synapses, some of which might not be stimulated. In other words, given the position of the stimulation electrode distal from the cell body layer, the fraction of activated axons is likely higher at distal ROIs, compensating for their lower Pr. This argument (from the authors!) renders the burst experiments moot (Figure 3—figure supplement 1), as the read-out is df/f (measured in large ROIs), which is mainly a function of the density of active axons. Thus, the burst response slope is expected to be steepest close to the tip of the stimulation electrode. Indeed, ROIs close to the cell body layer (80-100 μm from soma layer) showed no response at all in these experiments (Figure 3—figure supplement 1C)! The burst experiments should therefore be removed unless there is a possibility to normalize the data to the 1st pulse responses (as in PPR analysis).

3) Modeling (subsection “A realistic biophysical model explains the role of the release probability trend”), "with the Pr values distributed in accord with our data (Figure 1C)". According to the Materials and methods, you used the slope of the linear fit, which is not the same as the distribution of the measured Pr values (which are much more variable). This has to be clear in the text. In Figure 1C, some synapses have distances > 300 um, but the most distal synapses in the model seem to be just 150 μm from the soma. Is there a reason why you did not place synapses on the distal apical dendrite of the model? Given you put only proximal synapses in your model, was the average release probability and the PPR in the "Pr trend" simulation identical to the average from the measurements? The comparison of the different model scenarios only makes sense if the average Pr and facilitation was identical in all scenarios. Please show the distribution of synaptic properties for all scenarios. If the gradients really matter, I would also expect a comparison of the frequency response to proximal vs distal inputs (as in Grillo, 2018).

---

## [Author Response]

Essential revisions:The full reviews are included below to help clarify the concerns. Each of the concerns raised should be addressed, however many of these require simply textual clarifications. Key additional analyses required are:1) The reanalysis suggested by reviewer #3.2) An analysis of baseline noise distributions and of how apparent Ca+ and iGluSnFR "failure" signals vary with the site and intensity of stimulation, coupled to a more detailed quantitative argument as to why spillover affects the latter, but not the former measurements.

In brief, we have (a) added the PPR analysis suggested by reviewer #3 and corrected the related presentation errors (original analyses did use correct PPR formulae throughout); (b) added the signal-to-noise ratio data for all dendritic spines, (c) dealt systematically with all other reviewers' comments by revising and appending the text and illustrations, as further detailed below.

Reviewer #1:The authors use optical quantal analysis and iGluSnFR measurements of glutamate release at CA3-CA1 synapses to convincingly demonstrate a gradient in release probability that favors distal synapses, matching in many respects, the distance-dependent scaling demonstrated postsynaptically by Magee and others. This is an elegant study that reveals a new feature of hippocampal synaptic physiology. Although the net effect is modest (~15%) the authors also use a realistic biophysical model to demonstrate that the observed gradient improves information transfer to CA1 neurons.Figure 1: why are 1/3 of the synapses in C. missing in D. Were these synapses for which a second response was not clearly visible? Is this presumed due to saturation? Some brief description of the reason for exclusion should be provided.

Indeed, a proportion of synapses showed no detectable 0-1 responses whereas 1-1 responses could not be reliably resolved, either, from the decay of the first response, due to dye saturation. Whilst these data were excluded from the PPR analyses, we note that direct P_r_ measurements required the first-pulse scores only. The explanation has been added (subsection “Monitoring release probability with optical quantal analysis”, Figure 1D legend).

Reviewer #2:In this study, Jensen and colleagues use optical imaging of synaptic calcium signals and glutamate release and report that presynaptic release probability increases in CA1 pyramidal cell dendrites where more distal synapses have a higher release probability. These results complement earlier observations on the increase in postsynaptic efficacy in distal dendrites. While the findings presented in this manuscript address a fundamental question in synapse biology and are potentially important, further documentation is needed to ensure validity of these results and realize their full impact.1) The findings presented here contradict an earlier thorough study by Magee and colleagues that addressed the same question and reached opposite conclusions. I think the authors should discuss this work (Smith et al., 2003) and address the potential discrepancy.

This is a legitimate request. To gauge release probability along the dendritic tree, Smith et al., used an MK801-based approach (their Figure 8A) while recording NMDAR responses in a Mg-free solution and 2.5 mM Ca^2+^. While they detected no significant difference between proximal and distal synapses, the removal of Mg^2+^ might have had a significant, possibly heterogeneous, effect on P_r_ across synaptic populations. In addition, the MK801-based approach is sensitive to the extent of local extrasynaptic glutamate escape (spillover) (Scimemi et al., 2004), which could bias comparative analyses of P_r_.

Intriguingly, Smith et al' s own data show 15-50% lower average PPR values (consistently over 25-200 ms paired-pulse intervals) in distal compared to proximal synapses (their Figure 8Biii). Although their dataset did not reach statistical significance at n = 6, it clearly pointed to the trend reported by us. We have added the corresponding discussion (subsection “Monitoring release probability with optical quantal analysis”).

2) It is important to document baseline noise distributions for calcium indicator as well as glutamate probe measurements. It is critical to demonstrate the reliability of success/failure classification in individual recordings. What is the variation in response amplitudes? How do they compare to baseline fluorescence fluctuations in the same trace?

In our hands, the distinction between Ca^2+^ signal failures and successes, at least in response to the first stimulus, was unambiguous, as shown by some detailed analysis in our previous work (Sylantiev et al., 2013; Boddum et al., 2016). To follow the Reviewer's request, here we have measured the first-response amplitudes of successes and failures (over the same sampling interval post-pulse) in all trials throughout; the plot of average values (per spine, mean ± SD) has been added to Figure 1—figure supplement 1D. These data show that the success amplitude was consistently above 2SD of the failure signal fluctuations, and that there were no distance-dependent trends. The text has been expanded accordingly (subsection “Monitoring release probability with optical quantal analysis”).

iGluSnFR and OQA: Our previous work detailed a robust OQA approach for the iGluSnFR signals recorded in a single axon in response to action potentials elicited in one presynaptic cell (Jensen et al., 2017; 2019). However, attempts to employ a similar method under local extracellular stimulation turned out to be much less straightforward because in most cases it was not possible to separate out a glutamate spillover signal originating from neighbouring synapses. There are good reasons for that. In the hippocampal neuropil, released glutamate molecules can travel for up to ~2 μm (e.g., Armbuster et al., 2020) whereas individual synapses are likely to have >60 neighbours within the 2 μm proximity radius (sphere volume of ~33 μm^3^ at synaptic density of ~2 μm^-2^). Furthermore, 2PE imaging integrates fluorescent signal collected over an ~1 μm thick focal plane. Thus, local extracellular stimulation is likely to generate a glutamate spillover signal even if it excites only a small fraction of local axons. The iGluSnFR signals induced by local extracellular stimuli tend to report, therefore, release activity of a small group of local synapses, which makes this approach usable for PPR measurements rather than for direct P_r_ estimates. We have added the corresponding discussion (subsection “Monitoring glutamate release at CA3-CA1 synapses with an optical sensor”).

3) As the senior author and colleagues proposed 14 years ago PPR can be release-independent (Volynski et al., 2006) and therefore makes a poor predictor of release probability. Given the authors have near single synapse resolution, it is surprising that they heavily rely on PPR measure for their arguments. While paired pulse stimulation is a good tool to reduce the bias towards potential high Pr synapses, failure analysis (first response success probability) should be more emphasized over paired pulse ratios. It is worrisome that iGluSnFR experiments solely rely on PPR measure. The authors indicate that they cannot detect failures due to spill over contribution to their measurements. Wouldn't spill-over be more of a concern for calcium signals as they rely on NMDA receptors? Given this group's track record in the field, I would have expected more insight into this issue. I think the authors should at least make sure that this is not a simple technical issue (e.g. location and intensity of stimulation etc.).

Indeed, in the past we were able to detect a release-independent component of PPR at paired-pulse intervals of 5 ms (Volynski et al., 2006), but it has long been argued that such phenomena fade away at 50 ms intervals. Nonetheless, we have added an alternative PPR measure, which was suggested by reviewer #3 and which deals precisely with the release-independent probability of the second release (Figure 1E). This addition had no effect on our conclusions.

As explained above, 2PE iGluSnFR imaging under extracellular stimulation is likely to pick up glutamate spillover signals originating at numerous neighbouring synapses. However, the overwhelming numbers of high-affinity glutamate transporters (with or without iGluSnFR present) normally ensure that such spillover can produce only a tiny NMDAR current at neighbouring synapses (e.g., Armbuster et al., 2020). Given the classical non-linear and saturating (S-shaped) sensitivity of postsynaptic Ca^2+^ fluorescence to NMDAR current, spillover should have little effect on OQA readout near the bottom (release failure) and near the top (release success) of the scale. Indeed, multiple studies involving classical OQA or single-synapse LTP induction have indicated that, under suitable protocols, postsynaptic Ca^2+^ signals elicited either by local afferent stimuli or by glutamate uncaging, do not spread to neighbouring spines. We have added the corresponding explanations (subsection “Assessing Pr using optical quantal analysis and paired-pulse ratios”).

Reviewer #3:I am not able to assess the manuscript in its current form, as I have doubts about the extraction of quantal parameters from the fluorescence traces. The description in the Materials and methods does not seem to match the examples provided. I suggest re-analysis of the data (not just of the example traces!).In the Materials and methods, P2 is defined as "the sum of responses to the second stimulus + responses to both stimuli/total number of trials". There are 4 possible outcomes of paired pulse stimulation: double failure (0,0), single responses (1,0 and 0,1) and double responses (1,1). The author's definition sounds like P2 = [(0,1) + (1,1)] / [(0,0) + (1,0) + (0,1) + (1,1)], which makes sense.Looking at the example traces shown in Figure 1B and Figure 1—figure supplement 1C, however, the given values of P2 correspond to [(1,0) + (0,1) + (1,1)] / [(0,0) + (1,0) + (0,1) + (1,1)], which is actually a different quantity, (P1+P2). The resulting calculated PPRs are thus not correct.Given the SNR of the (best) examples, it seems very risky to try to distinguish between (1,0) and (1,1). My recommendation would be to define P2 = (0,1) / [(0,0) + (0,1)], which can be reliably extracted. Unfortunately, this requires complete re-analysis of all raw data and may change results and conclusions.

We fully appreciate this comment and apologise for the technical glitch: the P_r_ values in question were left in the figures from the previous (discarded) example traces – the replacement of labelling was overlooked, which has now been corrected. However, all the original PPR analyses were carried out in accord with the correct formula where P2 = {(0,1) + (1,1)} / (number of trials). In fact, all the success-failure scores and all related data, for all spines, could be found in the Source data file supplied with the manuscript (Figure 1—source data 1). We do appreciate the suggestion to use an alternative / adjusted P2 definition and have added these data to Figure 1E and illustrations. As expected, both the original PPR and adjusted PPR values provided qualitatively identical conclusions.

We note that the distinction between Ca^2+^ signal failures and successes, at least in response to the first stimulus, was unambiguous, as shown by some detailed analysis in our previous work (Sylantiev et al., 2013; Boddum et al., 2016). To provide a quantitative reference, we have measured the first-response amplitudes of successes and failures (over the same sampling interval post-pulse) in all trials throughout. The plot of average values (per spine, mean ± SD) has been added to Figure 1—figure supplement 1D. These data show that the success amplitude was consistently above 2SD of the failure-signal fluctuations, and that there were no distance-dependent trends. The text has been expanded accordingly (subsection “Monitoring release probability with optical quantal analysis”).

[Editors' note: further revisions were suggested prior to acceptance, as described below.]

Essential revisions:1) Reviewer 3 correctly points out that the effect size for the central observation is small. This does not necessarily preclude publication, but this point should be made clearer to the reader both by including the correlation coefficients in the figures and adding consideration of this issue to the Discussion.

While we appreciate this comment, the interpretation of Pearson's r depends on the context. In our case, the unexplained variance (1-r^2^) reflects inherent biological variability of P_r_ among individual synapses, which is a well acknowledged phenomenon. The central effect we see is not small, it is rather large (P_r_ changes from ~0.2 at proximal to ~0.5 at distal synapses), but it is the large P_r_ variability that leads to a relatively low r. The reviewer correctly points out that low r values imply poor predictability of the regression trend based on an individual specimen: thus, high Pearson's r could be critical in clinical studies that seek to classify an individual's condition. But this is irrelevant to our quest because we are not interested in the action of individual synapses. As nerve cells are incessantly bombarded by synaptic discharges, our focus is on the overall trend (that transpires over virtually unlimited sampling), rather than on the symmetric variability / noise around this trend. In this context, high P_r_ variability is effectively cancelled out as we observe the average trend: the latter, in turn, has a significant effect on signal integration, as shown by our simulations.

To illustrate our point further, here we pooled and averaged all individual P_r_ values over the respective 25 μm distance bins. This narrowed down the P_r_ scatter along the regression line and thus increased Pearson's r to a 'strong' ~0.80. This synapsepooling exercise is somewhat representative of the case of multi-synaptic responses, as our iGluSnFR recordings or in Grillo et al., 2018. Judging by Pearson's r, the effect here appears much 'stronger' than we had with individual P_r_ values (r ~0.41), but in reality, it represents exactly the same P_r_ data set. We would therefore argue against overinterpreting Pearson's r as an indicator of the effect 'strength' of ‘weakness', in the present context.

We have expanded our explanations as a discussion point (subsection “Monitoring release probability with optical quantal analysis”, subsection “Possible implications for dendritic signal integration”) and revised figure labelling as suggested.

2) The analysis of the burst experiments should be revised along the lines suggested below, or these experiments should be removed or the concern raised adequately rebutted.

The corresponding reviewer's critique reads *"*Thus, the burst response slope is expected to be steepest close to the tip of the stimulation electrode. Indeed, ROIs close to the cell body layer (80-100 μm from soma layer) showed no response at all in these experiments (Figure 3—figure supplement 1C)!…".

There must have been some confusion here. The data in question (Figure 3—figure supplement 1) deal with the regression slope between first - fifth ΔF/F response peaks, which we employed as a crude short-term plasticity indicator during the burst. This slope indicator (in %ΔF/F s^-1^ units) has nothing to do with the absolute ΔF/F amplitudes, hence little to do with the synaptic density or electrode position. It is this regression slope (between first - fifth response peaks) that was close to zero in Figure 3—figure supplement 1C whereas the ΔF/F amplitudes were equally large at proximal and distal sites, as shown in Figure 3D. We have clarified this further (subsection “Monitoring glutamate release at CA3-CA1 synapses with an optical sensor”; Figure 3—figure supplement 1C legend).

3) Whether or not the average Pr and facilitation were kept constant for the different modeling scenarios should be clarified. If this is not the case, it will be necessary to revise the modeling portion of the study, or to make a convincing argument as to why this is not required in order to demonstrate the functional significance of the observed trend.

We can only apologise for not making it abundantly clear that the average P_r_ = 0.36 has been kept constant throughout simulations: comparing cases with different overall synaptic inputs would indeed be a pointless exercise. The clarification has been added (subsection “A realistic biophysical model explains the role of the release probability trend”, subsection “NEURON modelling”).

Reviewer #3:1) The discussion of correlations is focused on their p-values (which indicate that the slope of a linear regression is not zero). Biologically more interesting are the correlation coefficients, which indicate (after squaring) that only a small part of the total variability in Pr is determined by synapse position. Correlations with R squared = 0.18 (Figure 1C) or 0.14 (Figure 2A) are considered "very weak", which is not necessarily what you expect given the title of the manuscript. As all data points are shown, readers will form their own opinion and realize that synapse position has only a weak impact on presynaptic properties. Still, I recommend showing the numerical value of r (or r squared) on all panels with linear fits (and relegate "n" to the figure legends).

While we appreciate this comment, the interpretation of Pearson's r depends on the context. In our case, the unexplained variance (1-r^2^) reflects inherent biological variability of P_r_ among individual synapses, which is a well acknowledged phenomenon. The central effect we see is not small, it is rather large (P_r_ changes from ~0.2 at proximal to ~0.5 at distal synapses), but it is the large P_r_ variability that leads to a relatively low r. The reviewer correctly points out that low r values imply poor predictability of the regression trend based on an individual specimen: thus, high Pearson's r could be critical in clinical studies that seek to classify an individual's condition. But this is irrelevant to our quest because we are not interested in the action of individual synapses. As nerve cells are incessantly bombarded by synaptic discharges, our focus is on the overall trend (that transpires over virtually unlimited sampling), rather than on the symmetric variability / noise around this trend. In this context, high P_r_ variability is effectively cancelled out as we observe the average trend: the latter, in turn, has a significant effect on signal integration, as shown by our simulations.

To illustrate our point further, here we pooled and averaged all individual P_r_ values over the respective 25 μm distance bins. This narrowed down the P_r_ scatter along the regression line and thus increased Pearson's r to a 'strong' ~0.80. This synapsepooling exercise is somewhat representative of the case of multi-synaptic responses, as our iGluSnFR recordings or in Grillo et al., 2018. Judging by Pearson's r, the effect here appears much 'stronger' than we had with individual P_r_ values (r ~0.41), but in reality, it represents exactly the same P_r_ data set. See Author response image 1.

We would therefore argue against overinterpreting Pearson's r as an indicator of the effect 'strength' of ‘weakness', in the present context.

We have expanded our explanations as a discussion point (subsection “Monitoring release probability with optical quantal analysis”, subsection “Possible implications for dendritic signal integration”) and revised figure labelling as suggested.

2) Figure 3C shows strong facilitation at proximal synapses, linear behavior at distal synapses, which the authors take as supporting evidence that Pr is higher at distal synapses. However, the first pulse amplitude is similar (Figure 3D, averaging failures and successes), which the authors explain away by saying these measurements average glutamate from at least 60 synapses, some of which might not be stimulated. In other words, given the position of the stimulation electrode distal from the cell body layer, the fraction of activated axons is likely higher at distal ROIs, compensating for their lower Pr. This argument (from the authors!) renders the burst experiments moot (Figure 3 supplement 1), as the read-out is df/f (measured in large ROIs), which is mainly a function of the density of active axons.

We note here that the ΔF/F amplitude generated under extracellular stimulation is a poor predictor of P_r_ because of multiple, poorly-controlled factors affecting the optical glutamate spillover signal, such as: varied local synaptic density or inter-synaptic distance (e.g., a sharp increase in the occurrence of multi-synapse axonal boutons along the s. radiatum – s. lacunosum moleculare direction has been reported, Bloss et al., 2018), varied axonal density or axonal excitability under the stimulating electrode, varied expression ratio for iGluSnFR vs GLT-1 transporters that would have a complex effect on glutamate escape per se as well as on the detected optical signal, etc. These issues have been discussed in the text.

Thus, the burst response slope is expected to be steepest close to the tip of the stimulation electrode. Indeed, ROIs close to the cell body layer (80-100 μm from soma layer) showed no response at all in these experiments (Figure 3—figure supplement 1C)! The burst experiments should therefore be removed unless there is a possibility to normalize the data to the 1st pulse responses (as in PPR analysis).

There must have been some confusion here. The data in question (Figure 3—figure supplement 1) deal with the regression slope between first - fifth ΔF/F response peaks, which we employed as a crude short-term plasticity indicator during the burst. This slope indicator (in %ΔF/F s^-1^ units) has nothing to do with the absolute ΔF/F amplitudes, hence little to do with the synaptic density or electrode position. It is this regression slope (between first - fifth response peaks) that was close to zero in Figure 3—figure supplement 1C whereas the ΔF/F amplitudes were equally large at proximal and distal sites, as shown in Figure 3D. We have clarified this further (subsection “Monitoring glutamate release at CA3-CA1 synapses with an optical sensor”; Figure 3—figure supplement 1C legend).

3) Modeling (subsection “A realistic biophysical model explains the role of the release probability trend”), "with the Pr values distributed in accord with our data (Figure 1C)". According to the methods, you used the slope of the linear fit, which is not the same as the distribution of the measured Pr values (which are much more variable). This has to be clear in the text.

The reviewer is correct: we should have said 'regression trend' or the like. Clarified (subsection “A realistic biophysical model explains the role of the release probability trend”, subsection “NEURON modelling”).

In Figure 1C, some synapses have distances > 300 um, but the most distal synapses in the model seem to be just 150 μm from the soma. Is there a reason why you did not place synapses on the distal apical dendrite of the model? Given you put only proximal synapses in your model, was the average release probability and the PPR in the "Pr trend" simulation identical to the average from the measurements?

Figure 4A: We must admit that placing labelling or scale bars separately from the embedded images leads to trivial unexpected glitches during figure panel arrangements. Author response image 2 is the original NEURON rendering, with the original μm scale.

**Author response image 2. respfig2:** 

The synapses in the model were scattered as originally described. The scale bar in Figure 4A has therefore been corrected.

The comparison of the different model scenarios only makes sense if the average Pr and facilitation was identical in all scenarios.

We can only apologise for not making it abundantly clear that the average P_r_ = 0.36 has been kept constant throughout simulations: comparing cases with different overall synaptic inputs would indeed be a pointless exercise. The clarification has been added (subsection “A realistic biophysical model explains the role of the release probability trend”, subsection “NEURON modelling”).

Please show the distribution of synaptic properties for all scenarios. If the gradients really matter, I would also expect a comparison of the frequency response to proximal vs distal inputs (as in Grillo, 2018).

We have clarified further the setting of P_r_ in the Materials and methods. As for comparing the effects of proximal versus distal synapses with the Pr trend, this would mean that we simply compare synaptic groups with a stronger or weaker overall release efficacy. This reviewer has argued above that such comparisons are not very informative.